# Neural Basis of Brain Dysfunction Produced by Early Sleep Problems

**DOI:** 10.3390/brainsci6010005

**Published:** 2016-01-29

**Authors:** Jun Kohyama

**Affiliations:** Tokyo Bay Urayasu Ichikawa Medical Center, 3-4-32 Todaijima, Urayasu 279-0001, Japan; j-kohyama@jadecom.or.jp; Tel.: +81-47-351-3101

**Keywords:** sleep, serotonin, striatum, prefrontal cortex, cohort, marshmallow test

## Abstract

There is a wealth of evidence that disrupted sleep and circadian rhythms, which are common in modern society even during the early stages of life, have unfavorable effects on brain function. Altered brain function can cause problem behaviors later in life, such as truancy from or dropping out of school, quitting employment, and committing suicide. In this review, we discuss findings from several large cohort studies together with recent results of a cohort study using the marshmallow test, which was first introduced in the 1960s. This test assessed the ability of four-year-olds to delay gratification and showed how this ability correlated with success later in life. The role of the serotonergic system in sleep and how this role changes with age are also discussed. The serotonergic system is involved in reward processing and interactions with the dorsal striatum, ventral striatum, and the prefrontal cortex are thought to comprise the neural basis for behavioral patterns that are affected by the quantity, quality, and timing of sleep early in life.

## 1. Introduction

Sleep loss affects various brain functions in both adults and children [1] and is associated with behavioral, cognitive, and physical problems [2,3,4], as well as atypical early development [5]. Although shorter childhood sleep times were reported to be associated with higher adult body mass index values [6], long term effects of poor sleep and circadian rhythms on brain functions have not been fully understood. Recently, poor sleep and circadian rhythms during the early stages of life are hypothesized to produce changes in brain function that cause behavioral abnormalities later in life (truancy and dropping out of school, quitting employment, and committing suicide) [7]. However, the neural basis for these behavioral changes has not been fully determined; an involvement of serotonin (5-hydroxytryptamine (5-HT)), oxytocin, and the prefrontal cortex (PFC) has been proposed (Figure 1 [7]).

This review begins by briefly summarizing the effects of sleep on the brain. We then describe the studies that have shown unfavorable effects of sleep deterioration during the early stages of life on behavior. Finally, we discuss the role of 5-HT in changing brain functions affected by poor sleep and circadian rhythms, that may produce behavioral abnormalities later in life.

## 2. Sleep Affects the Structure and Function of the Brain

Closure of one eye in 4–5-week-old kittens decreased the number of neurons responding to photic stimulation in the primary visual cortex. This indicates that during a critical developmental period, monocular deprivation (MD) initiates rapid synaptic remodeling of neurons in the primary visual cortex, in favor of the open eye [8]. Frank *et al.* [9] went on to show that laterality induced by MD as described above was hindered by sleep deprivation, as well as enhanced by sleep. This suggested that sleep in the early stages of life plays a crucial role in the functionality of the developing brain.

**Figure 1 brainsci-06-00005-f001:**
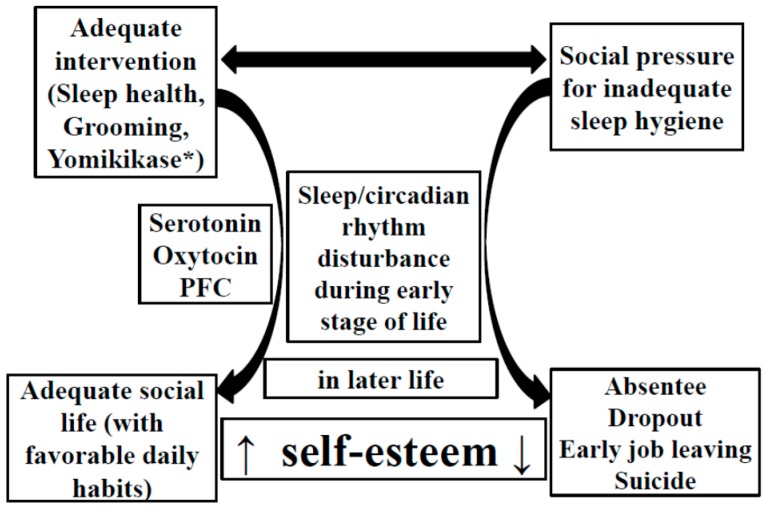
Proposed relationship between sleep hygiene, daily habits, neuronal elements, and self-esteem. Adapted from Kohyama [7] *Expert. Rev. Neurother.*
**2014**, *14*, 745. Copyright © 2014 with permission of Expert Review of Neurotherapeutics. * Yomikikase refers to a Japanese phrase, meaning to read a book to another person [10]. Yomikikase is reported to activate the limbic structures of children [11] and increase the frontal cerebral blood flow of the reader [12].

In one-month-old mice, the formation of dendritic spines differs continuously; spine formation and elimination is prominent during periods of wakefulness and sleep respectively, resulting in a net increase or decrease in spine density [13]. Yang and Gan [14] reported greater spine formation in three-week-old mice during the dark period, when mice are mostly awake, and more elimination of these protrusions during the light period, when mice are mostly asleep. These findings confirmed that, in young mice, the amount of sleep and wakefulness can affect the density of cortical synapses. The authors also reported that sleep after motor learning promoted the formation of postsynaptic dendritic spines on a subset of branches in individual layer-V pyramidal neurons and that neurons activated during motor task learning were reactivated during subsequent non-rapid eye movement sleep [15]. Disrupting this neuronal reactivation prevented branch-specific spine formation. These findings indicated that sleep promotes learning-dependent synapse formation and maintains dendritic branches, which contributes to the storage of motor memories.

Recently, it was proposed that sleep restores synaptic homeostasis after periods of wakefulness, during which time synapses are formed by learning [16]. In adult mice, spine turnover is limited and is not influenced by a state of sleep or wakefulness [13], suggesting that after adolescence, synaptic homeostasis is mediated primarily by changes in synaptic strength rather than number. The effect of sleep disturbance on the brain is difficult to study in humans, but white matter is found to be extensively altered in adult patients with obstructive sleep apnea [17]. Diffusion tensor imaging has shown that sleep disruption is associated with lower white matter integrity in adolescents [18]. Moreover, obstructive childhood sleep apnea has been associated with possible damage to the hippocampus and frontal cortex in addition to lower intelligence quotient and executive function [19]. These findings suggest that sleep disturbance can affect the morphology and function of the brain in humans.

## 3. Sleep Deterioration during the Early Stage of Life Has a Long-Term Negative Impact on Brain Function

Severe sleep problems in infancy, including bedtime struggles and difficulty falling asleep have been associated with the development of attention deficit hyperactivity disorder [20]. Nightmares, sleepwalking and trouble sleeping at four years of age were reported to predict behavioral and emotional problems during mid-adolescence [21]. Similar sleep problems between four and 19 years of age were reported to be related to attention problems, aggressive behaviors, anxiety, and depression later in life [22]. In a large twin study, sleep problems including frequent/early waking, nightmares, and resistance to sleep at three and four years of age predicted anxiety, conduct problems, and hyperactivity at seven years of age [23]. A recent nationwide longitudinal survey in Japan revealed that poor sleep schedules in toddlers predicted behavioral problems at primary-school age [24]. Compared with waking early, irregular or late morning waking at two years was more significantly associated with overly aggressive behavior at eight years of age (odds ratio (OR), 1.52; 95% confidence interval (CI), 1.04–2.22). Furthermore, compared with an early bedtime, an irregular or late bedtime at two years of age was more significantly associated with attention problems (OR, 1.62; 95% CI, 1.12–2.36) and aggressiveness (OR, 1.81; 95% CI, 1.19–2.77) at eight years of age [24]. These findings [23,24] indicated that unfavorable sleep and/or circadian habits during the early stages of life may lead to conduct problems or aggressiveness during the first years of elementary school. The recent increase in violent behaviors in elementary schools in Japan (>11,000 cases in 2014) [25] might be related to early sleep problems [7].

Several large cohort studies have investigated the importance of sleep. In Japan, the Toyama cohort study [26], which included more than 10,000 people born in 1989, revealed that later bedtimes, later waking times, short sleep durations, physical inactivity, skipping breakfast, irregular snacks, and frequent instant noodle consumption at three years of age were associated with increased risk of poor life quality in the first-year junior high school students. The same study also revealed a relationship between bedtimes at three years of age and those at 9–10 years of age [27]. More than half (56.8%) of children who went to bed before 9 pm at three years of age went to bed before 9:30 pm when they reached 9–10 years of age. Furthermore, 42.0% of children who went to bed later than 10 pm at 3 years of age went to bed after 10 pm when they reached 9–10 years of age. Thus, three-year-old children who were late sleepers tended to maintain their late bedtime habit at least until they were 9–10 years old, although environmental conditions such as family lifestyle might also have affected these habits.

The Dunedin Multidisciplinary Health and Development Study was conducted for more than 30 years and has provided much useful information regarding sleep [28]. After controlling for childhood internalizing problems, sex, and socioeconomic status, persistent sleep problems during childhood (ages five, seven, and nine years) predicted anxiety disorders in adulthood (ages 21 and 26 years; OR, 1.60; 95% CI, 1.05–2.45, *p* = 0.030) [29]. The Dunedin study [30] also revealed a relationship between sleep problems at 5, 7, and 9 years of age and neuropsychological function at 13 years of age. The mean level of family conflict at 7–15 years of age also predicted insomnia at 18 years of age after controlling for sex, socioeconomic status, sleep problems at nine years, and self-reported health [31]. However, it should be noted that these studies [27,28,29,30,31] did not provide a specific definition of “sleep problems”. Sleep problems were identified based on answers to general non-specific questions. The Dunedin cohort study also revealed the following personality continuities [32]: (1) under-controlled three-year-olds (resembling the Chess–Thomas difficult type [33], included children who were impulsive, restless, negativistic, distractible, and labile in their emotional responses.) grew up to be impulsive, unreliable, and antisocial, and had more social and work-related conflicts; (2) inhibited three-year-olds (resembling the Chess–Thomas slow-to-warm-up type [33], included children who were socially reticent, fearful, and easily upset by strangers.) were more likely to be unassertive and depressed, and had fewer sources of social support.

Greene *et al.* [34] reported an association between childhood sleeping difficulties and adulthood depression after adjusting for maternal depression and sleeping difficulty in a 30-year prospective study conducted between 1975 and 2005. The study revealed that severe, but not moderate sleeping difficulties, at five years of age were associated with an increased risk of depression at 34 years of age (severe: OR, 1.9; 95% CI, 1.2–3.2; moderate: OR, 1.1; 95% CI, 0.9–1.3).

Truancy and dropping out of school, quitting jobs rapidly, and committing suicide have already been identified as tragic consequences of early sleep problems [7]. We have reviewed the evidence that poor sleep and circadian rhythms were also associated with behavioral problems later in life, such as aggression, anxiety, depression, conduct problems, hyperactivity, insomnia, and the inability to pay adequate attention. Some of these problems can be treated by agents that increase 5-HT activity [35]. Impulsive and aggressive behaviors are known to be modulated by 5-HT signaling [36]. In the next section, we review the 5-HT system and its role in sleep.

## 4. Theories on the Development of the 5-HT System

The aminergic neurons of the brainstem and the midbrain have crucial roles in brain development [37]. 5-HT neurons are involved in the development of particular brain functions at critical ages and via specific receptors [38]. Axon pruning of 5-HT visual cortex neurons occurs during the first seven postnatal days in rats [39]. Segawa [40] postulated that aminergic systems are involved in the sleep-wake (S-W) cycle, and that the 5-HT system plays a role in the development of brain function.

According to Segawa [40], the circadian S-W cycle develops in the first four months of life and involves human interaction, accommodation to a novel environment, and functional lateralization of the cortex. By one year of age (the second phase), daytime sleep begins to occur only in the afternoon and atonia is restricted to rapid eye movement sleep. At the same time, synaptogenesis begins in the cortex and antigravity activity induces locomotion, which is involved in the development of the frontal cortex through activation of the dopaminergic (DA) system. By 4–5 years of age (the third epoch), the two-phase S-W rhythm is completed and daytime sleep stops. Additionally, brainstem and hypothalamic functions are synchronized and start to modulate functions of the frontal cortex by activating the non-motor basal ganglia-thalamo-cortical circuits. The 5-HT neurons involved in the development of the biphasic S-W cycle synchronize the biological rhythms generated by the brainstem and mesencephalon by 5 years of age. The striatum is innervated by both 5-HT and DA neurons. 5-HT neurons become involved in the development of the non-motor basal ganglia thalamo-cortical circuit as DA neurons develop. This circuit projects to the orbitofrontal cortex and the anterior cingulate area. These non-motor circuits are postulated to be involved in the following functions: dorsolateral prefrontal circuit (executive function, cognitive tasks), lateral orbitofrontal circuit (empathetic and socially appropriate responses), anterior cingulate circuit (motivated behavior, procedural learning via reinforcement signals from the ventral tegmental area and the substantia nigra pars compacta to diffuse areas of the basal ganglia and the cortex). Developing DA and 5-HT neurons innervate the non-motor basal ganglia thalamo-cortical circuits at 4–5 years of age and become involved in the specific cognitive functions of the frontal cortex controlled by these circuits. This provides a stable psychobehavioral background for the development of higher cortical function.

Segawa did not discuss the development of the 5-HT system after the third epoch [40]. Although age-dependent morphological and functional alterations of the 5-HT system in humans remain to be determined, Arita [41] compared mean urinary 5-HT concentrations at rest (in the morning) between age groups (young (3–4 years old), junior (4–5 years old) and senior (5–6 years old) children). The mean value decreased with increasing age. According to Arita [41], urinary 5-HT concentrations in senior children were similar to adults, which is compatible with findings that brain 5-HT concentrations are higher in infancy than in adulthood, and decrease to reach a stable level during the course of postnatal development. Arita [41] emphasized that 3–6 years after birth is a critical period for the postnatal development of 5-HT neurons. In addition, Arita [41] proposed three factors that activate the 5-HT system: rhythmic movements, solar irradiation, and grooming. Young [42] proposed four non-pharmacologic methods for increasing brain 5-HT: (1) food (tryptophan-enriched diet); (2) positive (happy) mood; (3) exposure to bright light; and (4) exercise.

The 5-HT system is activated during wakefulness, and its activity is reduced during non-rapid eye movement sleep and ceases during rapid eye movement sleep [43]. Contradictory to earlier theories, the activity of the 5-HT system is strongly related to sleep. During wakefulness, the 5-HT system is activated by exposure to morning light [44] and by rhythmic movements, such as gait, chewing, and respiration [41,42,43]. Unfavorable daily habits, such as poor physical activity (poor rhythmic movements) and waking late (reduced light exposure) fail to enhance 5-HT activity. According to the international classification of sleep disorders version 3 (ICSD-3) [45], insufficient sleep leads to fatigue and malaise, which naturally results in poor physical activity. In addition, irritability, concentration/attention deficits, and reduced vigilance are features of insufficient sleep syndrome. Insufficient sleep, which is often associated with late bed times [46], may result in reduced 5-HT activity through both poor physical activity and loss of exposure to morning light. Essentially, activation of the 5-HT system can be achieved by adequate sleep health, which promotes morning sunlight exposure and sufficient physical activity. Arita [41] illustrated means of increasing 5-HT system as Figure 2. Here, factors that modulate 5-HT system and the presumable roles of 5-HT are summarized in Figure 3 by modifying from Arita’s original figure (Figure 2) [41].

**Figure 2 brainsci-06-00005-f002:**
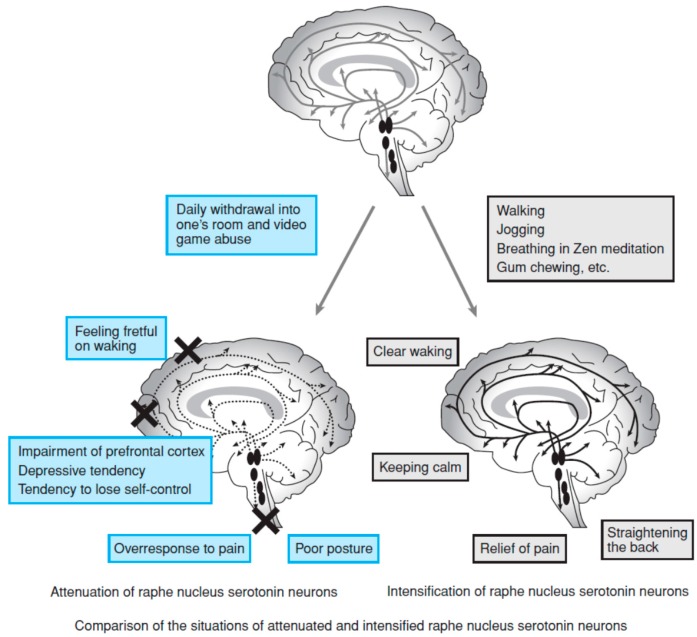
Adapted from Arita [41] *JMAJ*
**2009**, *52*, 184. Copyright © 2009 with permission from Japan Medical Association (the tiltle of original figure is “means of increasing serotonin”).

**Figure 3 brainsci-06-00005-f003:**
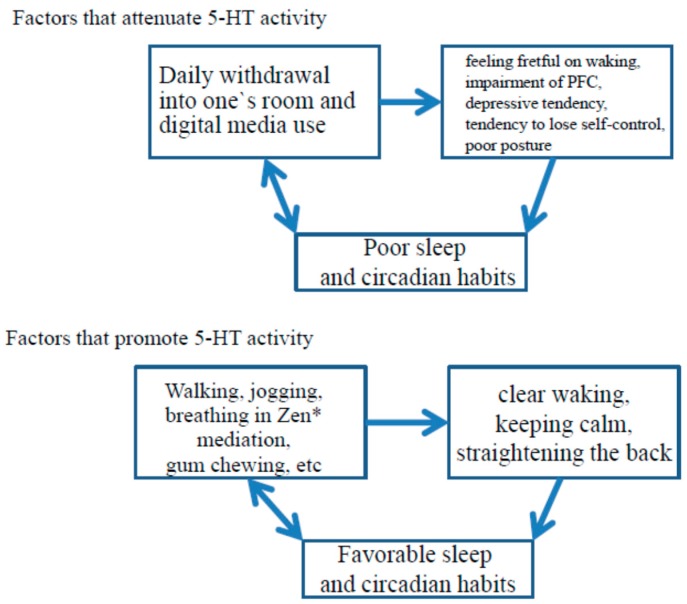
Factors that modulate 5-HT activity and the presumable roles of 5-HT (modified from Arita [41] (Figure 2)).

## 5. Marshmallow Test and Possible Involvement of the 5-HT System

The marshmallow test was developed by Mischel and colleagues [47]. Originally conducted in the 1960s to assess the ability to delay gratification, this test presented a group of four-year-olds with a choice between a small reward (one marshmallow), which they could have immediately, and a larger reward (two marshmallows), for which they had to wait 15 minutes [48]. About half the children chose to delay gratification. In numerous follow-up studies over a 40-year period, this test has been shown to have a surprisingly significant predictive validity for social, cognitive, and mental health later in life [49]. Four-year-old children who delayed gratification longer developed into more cognitively- and socially-competent adolescents, achieving higher scholastic performance, and coping better with frustration and stress [50]. According to Mischel [47], these processes involve the cool cognitive system, primarily centered in the PFC. Individuals who were less able to delay gratification consistently showed low self-control in their twenties and thirties and could not suppress responses to a happy face as well as those individuals who had the ability to delay gratification as children. According to Mischel, these processes are activated by the hot emotional system, closely related with the limbic system [47]. The results of the marshmallow test show that behavior at four years of age can predict the ability to self-regulate 40 years later [51].

According to Mischel [47], the PFC is crucial for future-oriented decisions and self-control (cool system). Even mild stress, if acute and uncontrollable, can cause a rapid and dramatic loss of prefrontal cognitive abilities [45]. Sleep deprivation decreases brain activity in the fronto-parietal attention network (PFC and intraparietal sulcus) in adult humans [52]. The medial PFC (paracingulate cortex) is also involved in developing a theory of mind, which occurs by five years of age [53]. The ventromedial PFC and subcortical areas (ventral tegmental area, nucleus accumbens and amygdala) are also sensitive to various types of rewards [54,55]. Dorsal and medial raphe nuclei, which contain many 5-HT neurons are also implicated in reward-directed behavior [54]. Casey *et al.* [45] revealed that children who were less able to delay gratification showed lower self-control in their forties. Casey *et al.* [45] also showed enhanced activation in the ventral striatum (hot system) in low delayers, and higher differentiation between no-go and go trials in the PFC of high delayers. The Dunedin study showed that specific personality traits were also continued from childhood through to adulthood [32], but the marshmallow study also related these observations to specific neuronal circuits.

The marshmallow test assessed the ability of a young child to resist an immediate reward in favor of a larger reward later on [50]. DA neurons are commonly referred to as “reward” neurons [56] and modulation of the DA reward system by 5-HT has received much attention [57]. The correlation between short-term reward seeking and ventral striatum activity was stronger when the level of 5-HT was lower (achieved by dietary tryptophan depletion), and a higher 5-HT level (achieved by dietary tryptophan loading) enhanced the correlation between long-term reward seeking and dorsal striatum activity [58]. Activity in the dorsal striatum also correlated with long-term reward predictions, while activity in the ventral striatum correlated more with short-term predictions. However, no significant differences were observed in the choice between immediate and delayed rewards at different dietary tryptophan levels [58], in agreement with a previous human study [59]. These authors later reported an increase in small reward choices and the rate of discounting of delayed rewards in the low-5-HT condition compared with the high-5-HT conditions [60]. In addition, dietary tryptophan depletion was associated with aggressive harvesting strategies during games, as well as disrupted use of the social norms, complete exhaustion of resources, and accumulation of fewer rewards than participants who were not tryptophan-depleted [61]. Therefore, a reduced ability to delay gratification in the marshmallow test may be linked to lower levels of 5-HT. 5-HT levels are affected by sleep during the first four years of life. It has been suggested that reward networks and circadian rhythms have evolved via a common mechanism [54]; therefore, there may be a link between early sleep habits and behavior later in life, although this remains to be determined.

## 6. 5-HT Activity, Sleep, and Development

Hypofunction of the 5-HT system in the PFC has been associated with suicide [62]. When people are sleep-deprived, they are unlikely to be active enough to produce sufficient 5-HT. Sleep loss decreases activity of the PFC [52], and decreased 5-HT activity, especially in the PFC, may play a role in suicidal behaviors [63]. A recent meta-analysis of data collected between 1966 and 2011 [64] supports the idea that sleep disturbances represent a risk factor for suicidal thoughts and behavior. The hot system [47] may be involved in suicidal thoughts and behavior. The Dunedin and Toyama studies mentioned three years as being important, the marshmallow test [47] focused on age four, Segawa [40] targeted ages 4–5, a theory of mind [53] emphasized age five, and Arita [41] paid attention to age 3–6 years. Unfortunately, we still have no data on the functional and/or morphological co-development of the 5-HT system, the striatum, and the PFC, although insufficient sleep syndrome is known to affect all ages and sexes [45].

To obtain an adequate social life, basic 5-HT activity is considered important at 4–5 [40] or 3–6 [41] years of age. Sufficient 5-HT activity can be achieved non-pharmacologically by exposure to morning sunlight and physical activity. Good sleep habits and circadian rhythms can help achieve this. The inability to delay gratification at four years of age cannot be reversed in adulthood, even by raised 5-HT levels; therefore, sufficient maturation of the 5-HT system might be crucial early in development for long-term reward seeking and this requires good sleep and circadian rhythms. T2-weighted imaging showed a different pattern of development in the dorsal and ventral striatum [65]. Good sleep and circadian rhythms during early stages of life promotes favorable development of the dorsal striatum involved in long-term reward seeking.

The sensitive period and the neural basis of self-esteem (Figure 1) remain to be determined. Mischel *et al.* [49] reported that the ability to delay gratification predicted less physical and verbal aggression, less bullying behavior, and higher self-worth and self-esteem, which underlines the importance of self-regulation strategies to overcome immediate temptations in children. These strategies may help the exertion of willpower later in life.

The relationship between hot and cool neural circuitry and self-esteem must be further investigated. Although a complete clarification of the neural basis of brain dysfunction produced by unfavorable sleep and circadian rhythms during early stages of life has not yet been accomplished, consistent progress in the field has helped us reach a more complete understanding.

## 7. Conclusions

Issues discussed in this review were added to Figure 1 (Figure 4).

**Figure 4 brainsci-06-00005-f004:**
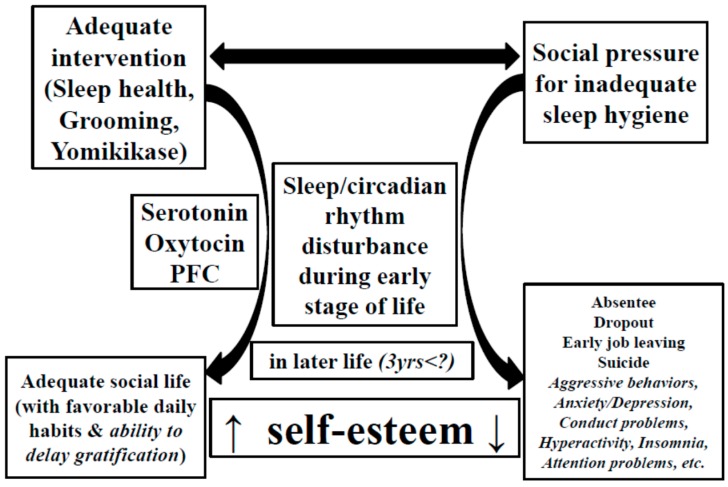
Proposed relationship between sleep hygiene, daily habits, neuronal elements, and self-esteem, modified from a previous review [7]. Expressions in italics were discussed in this review.

Several studies support the idea that poor sleep habits at young ages produce behavioral problems later on in life. These problems are likely to result from changes in the 5-HT system, the dorsal striatum, the ventral striatum, and the PFC, which affect brain development and related behaviors as the children grow older.

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
