# Peer review of "Neural Basis of Brain Dysfunction Produced by Early Sleep Problems"

_brainsci, 2016, doi:10.3390/brainsci6010005_

Round 1

Reviewer 1 Report

The topic is interesting. My comment is that the sub-chapters # 2 and # 3 must be extended by including other recent researches in the field

Author Response

Rev. 1.

Thank you for giving me useful comments.  I added 3 references (refs. 5, 16, 34) in accordance with the recommendation made by reviewer 1.

Reviewer 2 Report

This brief review, “Neural basis of brain dysfunction produced by early sleep problems,” provides an integration of several lines of research suggesting that disrupted sleep health (e.g., behavioral sleep problems, short sleep duration, late sleep timing) and circadian rhythms may be pathways underling links between early childhood neurophysiology and the later emergence of poor health and social outcomes. Strengths of this review include: i) a comprehensive account of observational study findings that establishes a rational for understanding basic mechanisms; ii) the integration of human and pre-clinical research findings; and iii) a proposed theoretical framework.  The following constructive concerns are presented with the intent to strengthen the manuscript.  

Comments:

1.    The author does not provide a review of the literature that establishes that poor sleep health is prevalent in early childhood – not just in Japan as described in the author’s 2014 review – but more generally.  This revision could be obtained in just a few sentences.

2.    In this reviewer’s opinion, some parts of the review are too colloquial for a scientific review, and these aspects should be revised and/or deleted.   For example, page 5, the paragraph starting at line 180 and the paragraph starting at line 188 include several historical quotations that are superfluous.  Although interesting, these paragraphs are distracting and do not add anything of sustenance to the review.

3.    There are a number of typos, thus, closer review of the paper is needed.

4.    The paper would benefit greatly from the comments/edits of someone for whom English is their primary language.

5.    The proposed framework is not sufficiently described (Pages 9-10).  For example, Yomikikase is not a common term and is not defined.  Also, the quote by Mischel is not warranted in this section.  Instead, the author should walk the reader through the proposals made by this framework.

6.    Cause/effect language that is not appropriate is used several times (e.g., Page 3, line 97).  The author needs to closely review the paper and delineate between causation and association.

7.    Although critical periods are established in animals, this term is not appropriate with regard to humans.  Sensitive period is the term of choice in the developmental science field.

8.    Page 2, lines 54-55.  Delete the final sentence stating what the author will do in the future.

Author Response

Rev. 2

This brief review, “Neural basis of brain dysfunction produced by early sleep problems,” provides an integration of several lines of research suggesting that disrupted sleep health (e.g., behavioral sleep problems, short sleep duration, late sleep timing) and circadian rhythms may be pathways underling links between early childhood neurophysiology and the later emergence of poor health and social outcomes. Strengths of this review include: i) a comprehensive account of observational study findings that establishes a rational for understanding basic mechanisms; ii) the integration of human and pre-clinical research findings; and iii) a proposed theoretical framework.  The following constructive concerns are presented with the intent to strengthen the manuscript.  

Thank you giving me positive comments.

Comments:1.    The author does not provide a review of the literature that establishes that poor sleep health is prevalent in early childhood – not just in Japan as described in the author’s 2014 review – but more generally.  This revision could be obtained in just a few sentences.

I added one sentence before the former introduction according to this recommendation.

2.    In this reviewer’s opinion, some parts of the review are too colloquial for a scientific review, and these aspects should be revised and/or deleted.   For example, page 5, the paragraph starting at line 180 and the paragraph starting at line 188 include several historical quotations that are superfluous.  Although interesting, these paragraphs are distracting and do not add anything of sustenance to the review.

Descriptions indicated by this comments were deleted. 

3.    There are a number of typos, thus, closer review of the paper is needed.

4.    The paper would benefit greatly from the comments/edits of someone for whom English is their primary language.

Thank you for giving me useful comments. The whole manuscript has been reviewed by native speakers.

5.    The proposed framework is not sufficiently described (Pages 9-10).  For example, Yomikikase is not a common term and is not defined.  Also, the quote by Mischel is not warranted in this section.  Instead, the author should walk the reader through the proposals made by this framework.

Thank you for giving me valuable comments.  According to this comment, Figure 1 was moved to the first section (introduction) of the manuscript. In addition, I tried to explain the role of Mischel study at p7 (The last part of the second column). Explanation on Yomikikase was added in the figure 1 legend. 

6.    Cause/effect language that is not appropriate is used several times (e.g., Page 3, line 97).  The author needs to closely review the paper and delineate between causation and association.

Thank you indicating my mistake.  I altered the expression.

7.    Although critical periods are established in animals, this term is not appropriate with regard to humans.  Sensitive period is the term of choice in the developmental science field.

According to this comments, I ceased to use the term critical period in humans.

8.    Page 2, lines 54-55.  Delete the final sentence stating what the author will do in the future.

According to this comment, I deleted the sentence. 

Reviewer 3 Report

Here the author presents an interesting review on how sleep early in life may scaffold (or conversely disrupt) serotonogic brain function (indexed in particular by a classic behavioral paradigm—the marshmellow delay task. This topic is timely and intresting, as the connection between sleep and pyschiatric conditions involving serotonergic dysfunction is actively under deabte. While the author includes many studies of interest the paper suffers from an overall lack of clear organization. For example, the paper opens with a traditional review of sleep-dependent brain plasticity early in life, before transitioning to discuss a single task (e.g., the marshmellow task) which indexes reward sensitivity and delayed gratification before proposing a neurobiological model of 5-HT which is not adequately explored in terms of hte neurocheistry of sleep. I do believe however that an adequate argument can be made for the impact of sleep loss on the serotonergic system (e.g,. depression) yet such instances are only mentioned briefly. While I outline several specific points below I believe this review could be strenghtend by a re-organization in at least three main ways: (1) Introducing the overall  schematic figure early rather than at the end, and using it to organize and order the sections, outlining each piece of the model before moving on; (2) Provide Figures illustrating the major data presnted; and (3) keeping the focus on sleep, whereby currently a large portion of the manuscript does not concern sleep.

Specific comments: 

1) The focus on Seratonin/oxytocin/and the PFC early in the first paragraph of the introduction is confusing as these studies do not relate to sleep. Can the author link these more clearly to the sleep changes laid out above?

2) Section 2 should be re-organized. Currently the author presents evidence from the Frank lab discussing monocular dominance plasticity and the impact of sleep, before tansitioning to OSA in childhood and then MRI studies in human adolescnece showing sleep as predictive of WM integrity, before returning to rodent models and discussing use-dependent changes in synaptic structure (e.g., spines). I belive the author is tyring to link together evidence that the structural morphology of hte brain is sensitive to hte history of sleep, particularly during early development. If so, this thesis should be made clearer at the beginning of this section. I suggest then that the author re-organizes this section with clear distinction between human and rodent work, or between different life-stages, for example.

3) The statement: "Compared with morningness, eveningness is known to cause behavioral, cognitive, and physical problems [33-35]." (line 97, pg 3)

Should be reconsidered. What I believe the author intends to demosntrate is that excessive owl-types when performing out of phase with their internal rhythm may suffer behavioral consequences. This is a "social jetlag" issue. I do not feel there is adequate evidence that eveningess "casuses" behavioral/cognitive or physical problems.

4) Many of the cohort studies (references 7-13) were conducted by the same group. Are these studies the same participants at different waves? The author presents them as if they are independent? 

5) In general Section 3 could use a restructing as well. The author covers many "types" of evidence, including circaidan phase preference, sleep schedules (e.g., bedtime), sleep hyigine, together with both behavioral and pyshicatric outcomes. It would be beneficial to resturcture this section so that circadian and sleep phenotypes are discussed separately. Also, it would be helpful to split the evidence based on type of outcome being measured (e.g., psychiatric morbidity, cognitive perforance, sleep perse).

6) While I understand the author's use of colloquial quotes in Section 4 to underscore the historical view of "nature vs nuture" the use of Bilbilcal and cultural statments as evidence of "tells us nature prevails over nurture" is inappropriate. We certainly have an abundance of scientific inquiry for how nurture can overcome genetic background. This statement is too rigid on its own, and the quotes are inappropriate for a neuroscience report.

7) Upon introduction of hte marshmellow delayed gratification experiments, it is unclear how the author intends to use this to relate back to sleep. The sleep evidence presented thus far do not point towards delay discoutning / reward-sensitivity / impulsivity. If this is the focus the author intends, perhaps revising the leading in sections to cover the role of sleep on impulsvity would be prudent.

8) As in examples above "It should be true that a part of someone’s personality, which is essentially the expression of brain activity, is established during the early stages of life, and that one component of a personality, expressed for example by marshmallow test, may involve 5-HT activity." is too rigid a statement, especially without reference.

9) Because section 5 is so focused on Serotonin's effect on delay discounting vis-a-vis the marshmellow test, it is unclear how it fits in the greater picture of the review on sleep.

10) The final discussion on 5-HT, the PFC and Sleep is well conceived in terms of integrating the findings, yet because hte prior the prior sectiosn are poorly organized it is unclear to the reader how each piece of data is placed togther. See opening comments on how to best restructure the manuscript.

Author Response

Rev. 3

Here the author presents an interesting review on how sleep early in life may scaffold (or conversely disrupt) serotonogic brain function (indexed in particular by a classic behavioral paradigm—the marshmellow delay task. This topic is timely and intresting, as the connection between sleep and pyschiatric conditions involving serotonergic dysfunction is actively under deabte. While the author includes many studies of interest the paper suffers from an overall lack of clear organization. For example, the paper opens with a traditional review of sleep-dependent brain plasticity early in life, before transitioning to discuss a single task (e.g., the marshmellow task) which indexes reward sensitivity and delayed gratification before proposing a neurobiological model of 5-HT which is not adequately explored in terms of hte neurocheistry of sleep. I do believe however that an adequate argument can be made for the impact of sleep loss on the serotonergic system (e.g,. depression) yet such instances are only mentioned briefly. While I outline several specific points below I believe this review could be strenghtend by a re-organization in at least three main ways: (1) Introducing the overall schematic figure early rather than at the end, and using it to organize and order the sections, outlining each piece of the model before moving on; (2) Provide Figures illustrating the major data presnted; and (3) keeping the focus on sleep, whereby currently a large portion of the manuscript does not concern sleep.

Thank you giving me useful comments.  According to these comments, the figure 1 was moved to the first section, the figure 2 was added to explain our hypothesis, I tried to focus on sleep, and I described the presumable relationship between marshmallow test and 5-HT system.

Specific comments: 

1) The focus on Seratonin/oxytocin/and the PFC early in the first paragraph of the introduction is confusing as these studies do not relate to sleep. Can the author link these more clearly to the sleep changes laid out above?

According to this comments, the indicated descriptions were deleted.

2) Section 2 should be re-organized. Currently the author presents evidence from the Frank lab discussing monocular dominance plasticity and the impact of sleep, before tansitioning to OSA in childhood and then MRI studies in human adolescnece showing sleep as predictive of WM integrity, before returning to rodent models and discussing use-dependent changes in synaptic structure (e.g., spines). I belive the author is tyring to link together evidence that the structural morphology of hte brain is sensitive to hte history of sleep, particularly during early development. If so, this thesis should be made clearer at the beginning of this section. I suggest then that the author re-organizes this section with clear distinction between human and rodent work, or between different life-stages, for example. 

I completely reorganize the second section in accordance with this comment.

3) The statement: "Compared with morningness, eveningness is known to cause behavioral, cognitive, and physical problems [33-35]." (line 97, pg 3)Should be reconsidered. What I believe the author intends to demosntrate is that excessive owl-types when performing out of phase with their internal rhythm may suffer behavioral consequences. This is a "social jetlag" issue. I do not feel there is adequate evidence that eveningess "casuses" behavioral/cognitive or physical problems. 

According to this comments, I deleted the use of term “cause” in this sentence.

4) Many of the cohort studies (references 7-13) were conducted by the same group. Are these studies the same participants at different waves? The author presents them as if they are independent?   

I tried to clarify the data origin according to this comment.

5) In general Section 3 could use a restructing as well. The author covers many "types" of evidence, including circaidan phase preference, sleep schedules (e.g., bedtime), sleep hyigine, together with both behavioral and pyshicatric outcomes. It would be beneficial to resturcture this section so that circadian and sleep phenotypes are discussed separately. Also, it would be helpful to split the evidence based on type of outcome being measured (e.g., psychiatric morbidity, cognitive perforance, sleep perse).  

Thank you for giving me valuable comment. According to this comment, section 3 was reconstructed.

6) While I understand the author's use of colloquial quotes in Section 4 to underscore the historical view of "nature vs nuture" the use of Bilbilcal and cultural statments as evidence of "tells us nature prevails over nurture" is inappropriate. We certainly have an abundance of scientific inquiry for how nurture can overcome genetic background. This statement is too rigid on its own, and the quotes are inappropriate for a neuroscience report.   

According to this comment, indicated descriptions were deleted.

7) Upon introduction of hte marshmellow delayed gratification experiments, it is unclear how the author intends to use this to relate back to sleep. The sleep evidence presented thus far do not point towards delay discoutning / reward-sensitivity / impulsivity. If this is the focus the author intends, perhaps revising the leading in sections to cover the role of sleep on impulsvity would be prudent.

Thank you for giving me useful comment.  According to this comment, figure 2 was added, and an explanation for the importance of marshmallow study was added (p7, the last part of the second column).

8) As in examples above "It should be true that a part of someone’s personality, which is essentially the expression of brain activity, is established during the early stages of life, and that one component of a personality, expressed for example by marshmallow test, may involve 5-HT activity." is too rigid a statement, especially without reference.  

The indicated sentence was deleted in accordance with this comment.

9) Because section 5 is so focused on Serotonin's effect on delay discounting vis-a-vis the marshmellow test, it is unclear how it fits in the greater picture of the review on sleep.  

I hope figure 2 would be helpful to show the relation between sleep and 5-HT system. 

10) The final discussion on 5-HT, the PFC and Sleep is well conceived in terms of integrating the findings, yet because hte prior the prior sectiosn are poorly organized it is unclear to the reader how each piece of data is placed togther. See opening comments on how to best restructure the manuscript.

According to this comments, the organization of the manuscript was altered in the revised version. I hope this reorganization would be helpful to understand this manuscript.

Round 2

Reviewer 2 Report

The author has made changes based upon the concerns of reviewers, and the manuscript is strengthened. Still, some of the language is awkward and grammatically incorrect. I highlight a few examples and provide suggestions. Page 1, Abstract: “There is a wealth of evidence that the disrupted sleep and circadian rhythm, which is common in modern society even during the early stages of life, has unfavorable effects on brain function.” Although there is a central circadian clock, there is not just one circadian rhythm – many rhythms exist. Suggestion: change to circadian rhythms. This should be revised across the entire manuscript. Also, “…the disrupted sleep and circadian rhythm, which is…” suggest that the author thinks that these represent a singular process. Suggested change: “…evidence that disrupted sleep and circadian rhythms, which are common…” Page 1, Abstract: “playing truant” should be changed to truancy throughout the entire paper. Page 4: “…in first-year junior high school students.” Suggested change: “…in the first year of junior high school.” Page 4: “The Dunedin 118 cohort study also revealed the following personality continuities : under-controlled 3-year-olds.” I don’t know how to define under-controlled. Do you mean poorly regulated? Page 4: “We have reviewed the evidence that poor sleep and circadian rhythm also cause behavioral problems.” I think you mean are associated with or predict – not cause. Page 5: “Factors that modulate the 5-HT system and the presumable roles of 5-HT are summarized in Fig. 2.” How is video game abuse relevant to early childhood? More relevant in this model may be digital media use. Page 8: “Unfortunately, we still have no data on the functional and/or morphological development of the 5-HT system, the striatum, and the PFC.” I do believe that we have some knowledge about the development of each of these systems; however, perhaps you were suggesting that we don’t have data on the co-development of them. Please be circumspect with this statement. Page 8: “The effect of age on sleep has not yet been elucidated, although it is known that poor sleep can occur at all ages and can also affect brain function at any age [43].” There is substantial knowledge of age-related and developmental changes in sleep across the lifespan, including sleep behavior and the EEG. This statement should be deleted or revised. Also, the following citation is inappropriate in support of this statement: American Academy of Sleep Medicine. The International classification of sleep disorder (Third 378 edition). American Academy of Sleep Medicine, Westchester, 2014.

Author Response

Reviewer 2

The author has made changes based upon the concerns of reviewers, and the manuscript is strengthened.

Thanks.

Still, some of the language is awkward and grammatically incorrect. I highlight a few examples and provide suggestions.

Thanks.

Page 1, Abstract: “There is a wealth of evidence that the disrupted sleep and circadian rhythm, which is common in modern society even during the early stages of life, has unfavorable effects on brain function.” Although there is a central circadian clock, there is not just one circadian rhythm – many rhythms exist. Suggestion: change to circadian rhythms. This should be revised across the entire manuscript. Also, “…the disrupted sleep and circadian rhythm, which is…” suggest that the author thinks that these represent a singular process. Suggested change: “…evidence that disrupted sleep and circadian rhythms, which are common…”

Thanks.  I corrected the above points as reviewer 2 suggested.

Page 1, Abstract: “playing truant” should be changed to truancy throughout the entire paper.

Thanks.  I corrected the above points as reviewer 2 suggested.

Page 4: “…in first-year junior high school students.” Suggested change: “…in the first year of junior high school.”

Thanks.  I corrected the above point as reviewer 2 suggested.

Page 4: “The Dunedin 118 cohort study also revealed the following personality continuities : under-controlled 3-year-olds.” I don’t know how to define under-controlled. Do you mean poorly regulated?

Thanks.  I cited ref. 33 and added explanation on undercontrolled child as well as inhibited child. 

Page 4: “We have reviewed the evidence that poor sleep and circadian rhythm also cause behavioral problems.” I think you mean are associated with or predict – not cause.

Thanks.  I corrected the above point as reviewer 2 suggested.

Page 5: “Factors that modulate the 5-HT system and the presumable roles of 5-HT are summarized in Fig. 2.” How is video game abuse relevant to early childhood? More relevant in this model may be digital media use.

Thanks.  I corrected the above point as reviewer 2 suggested in Fig. 3.

Page 8: “Unfortunately, we still have no data on the functional and/or morphological development of the 5-HT system, the striatum, and the PFC.” I do believe that we have some knowledge about the development of each of these systems; however, perhaps you were suggesting that we don’t have data on the co-development of them. Please be circumspect with this statement.

Thanks.  I corrected the above point as reviewer 2 suggested.

Page 8: “The effect of age on sleep has not yet been elucidated, although it is known that poor sleep can occur at all ages and can also affect brain function at any age [43].” There is substantial knowledge of age-related and developmental changes in sleep across the lifespan, including sleep behavior and the EEG. This statement should be deleted or revised. Also, the following citation is inappropriate in support of this statement: American Academy of Sleep Medicine. The International classification of sleep disorder (Third 378 edition). American Academy of Sleep Medicine, Westchester, 2014. 

Thanks.  I deleted the following sentence “The effect of age on sleep has not yet been elucidated”, but I cited AASM [45] as follows; insufficient sleep syndrome is known to affect all ages and sexes [45].

In addition, I cited Ref 6 with the following description; Although shorter childhood sleep times were reported to be associated with higher adult body mass index values [6], long term effects of poor sleep and circadian rhythms on brain functions have not been fully understood.

Reviewer 3 Report

I would like to thank the author for being very receptive to both my, and the other Reviewer's, comments. In sum, the author presents a much improved manuscript that is more focused on the issues of sleep, the development of the serotonin system, and it's impact on behavior.

The majority of my concerns have been adequately addressed. 

One major concern remains. In the previous review I requested that more primary data be illustrated, or re-printed from their original sources. I still beleive this is a neccessary addition to the current manuscript. Given the breadth of data reviewed, I would still prefer for the major pieces of data to be presented figures throughout the manuscript, such that a reviewer naive to this literature may greatly benefit from seeing hte primary data.

Currently the author includes heuristic schemata (particularly Fig. 2) which greatly aid the reader in synthesizing the material—an improvement over the previous version; however, a depiction of key, primary data, re-printed here from their source, will, in my opinion, greatly strengthen the report.

If put together with the changes already present in this revison, I believe the author has produced a more tightly constructed and focused review of sleep neurochemistry and behavior throughout development.

Author Response

Reviewer 3

I would like to thank the author for being very receptive to both my, and the other Reviewer's, comments. In sum, the author presents a much improved manuscript that is more focused on the issues of sleep, the development of the serotonin system, and it's impact on behavior. The majority of my concerns have been adequately addressed. 

Thanks.

One major concern remains. In the previous review I requested that more primary data be illustrated, or re-printed from their original sources. I still beleive this is a neccessary addition to the current manuscript. Given the breadth of data reviewed, I would still prefer for the major pieces of data to be presented figures throughout the manuscript, such that a reviewer naive to this literature may greatly benefit from seeing hte primary data.

Thanks. According to your suggestion, I presented the original figure as Fig. 1.

Currently the author includes heuristic schemata (particularly Fig. 2) which greatly aid the reader in synthesizing the material—an improvement over the previous version; however, a depiction of key, primary data, re-printed here from their source, will, in my opinion, greatly strengthen the report.

Thanks. AS your suggested, I cited figure 3 from Arita’s original reference [41].

If put together with the changes already present in this revison, I believe the author has produced a more tightly constructed and focused review of sleep neurochemistry and behavior throughout development.

Thanks. I added Ref 6 with the following description; Although shorter childhood sleep times were reported to be associated with higher adult body mass index values [6], long term effects of poor sleep and circadian rhythms on brain functions have not been fully understood.

Round 3

Reviewer 2 Report

The author has addressed all concerns.